# Biomimetic Design and Topology Optimization of Discontinuous Carbon Fiber-Reinforced Composite Lattice Structures

**DOI:** 10.3390/biomimetics8020148

**Published:** 2023-04-06

**Authors:** Zhong Hu

**Affiliations:** Department of Mechanical Engineering, South Dakota State University, Brookings, SD 57007, USA; zhong.hu@sdstate.edu

**Keywords:** biomimetics, topology optimization (TO), computer modeling, discontinuous carbon fibers (DiCFs), carbon fiber-reinforced polymer composites (CFRPCs), lattice structure, cuttlefish bone, mechanical properties, lightweight

## Abstract

The ever-increasing requirements for structural performance drive the research and development of lighter, stronger, tougher, and multifunctional composite materials, especially, the lattice structures, heterogeneities, or hybrid compositions have attracted great interest from the materials research community. If it is pushed to the extreme, these concepts can consist of highly controlled lattice structures subject to biomimetic material design and topology optimization (TO). However, the strong coupling among the composition and the topology of the porous microstructure hinders the conventional trial-and-error approaches. In this work, discontinuous carbon fiber-reinforced polymer matrix composite materials were adopted for structural design. A three-dimensional (3D) periodic lattice block inspired by cuttlefish bone combined with computer modeling-based topology optimization was proposed. Through computer modeling, complex 3D periodic lattice blocks with various porosities were topologically optimized and realized, and the mechanical properties of the topology-optimized lattice structures were characterized by computer modeling. The results of this work were compared with other similar designs and experiments to validate the effectiveness of the proposed method. The proposed approach provides a design tool for more affordable and higher-performance structural materials.

## 1. Introduction

Using traditional materials and conventional trial-and-error approaches, it is difficult or impossible to improve both the physical and structural performances. The increasing demands for structural performance have been driving the development of lightweight multifunctional composites and structures [1,2,3,4,5]. Lattice structures, hybrid compositions, or heterogeneities often possess multifunctional properties that can lead to high performance structures and are of great interest to the materials research community, especially the aerospace industry. The increasing use of composite materials is partly due to their potential for improvement. Versatility of the composite materials can range from mechanical to electrical and thermal. The most widely used composite materials have a polymer matrix, which is generally a poor conductor. For example, enhanced conductivity can be achieved through carbon fibers (CFs) and/or carbon nanotubes (CNTs) reinforcement.

Currently, the strongest materials available for structural applications are small diameter fibers (5–40 μm). The Young’s modulus of CFs reaches 250 GPa, and the specific strength reaches 2500 kN·m/kg, which is 10 times that of steel. The Young’s modulus of CNTs is 4 to 5 times stronger than that of CFs. The reduced defect size increases the strength of the fibrous materials, while the orientation of crystalline domains, strong conjugated C-C or C-N bonds or closed-packed polymer chains greatly increase the stiffness and strength of fiber-reinforced polymer composites (FRPCs) [6].

Polymer performance continues to improve. Polybenzimidazole, poly-ether-ether-ketone, and polyetherimide exhibit relatively high stiffness and strength as well as high performance at high temperatures. The application of FRPCs as structural materials has grown over the past 50 years due to the combination of low density, high stiffness/strength/toughness of the composite materials with porous lattice structures. In such structures, the fibers are aligned in the directions of the fabricated truss. The shear, compression, bending, and impact behaviors of the composite truss cores with different topologies and porosity were studied [7,8]. At a given weight, the lattice structure is 10 times stiffer than the existing ultralight materials and exhibits superior compressive strength and can be used to develop new lightweight multifunctional structures [9].

Although the stiffness and strength are mainly reinforced by fibers, FRPCs can dissipate large amounts of impact energy before fracture and are, thus, flaw insensitive. The high toughness of structural composites is also a result of their layered structure, which leads to the energy dissipation at different length scales during deformation and produces tough materials from brittle components [10,11]. This behavior has also been reported in structural biomaterials due to their complex hierarchical structure [12]. Composite properties can be carefully controlled in terms of matrix (critical to overall structural performance), fiber volume fraction, and distribution of laminates or fibers to form the structures for each application.

Fiber-reinforced porous composites with micro lattice truss topologies have been shown to bridge the apparent gap between existing materials and unreachable material limits in the low-density region of the Ashby’s chart for materials selection [13,14,15]. These periodic spatial lattice structures are frequently used as reinforcement for extremely lightweight structures on the one hand, and for explosion-proof and multifunctional applications on the other [16,17]. Utilizing such porous composites has additional advantages such as high energy absorption and excellent thermal and acoustic insulation [18,19,20,21].

Porous composites reinforced by discontinuous fibers (DiFs) offer great flexibility in tailoring specific physical properties by controlling the composition, the microstructures of the constituents, and the fiber orientation during fabrication. Periodic porous composites consist of many identical base blocks. Manipulating the phase distribution and fiber orientation in these base blocks provides an effective way to design multifunctional structural composites [22,23,24,25,26]. One of the intensive applications of the extensive research on porous composite structures is the development of thin-walled energy-absorbing components in the automotive industry. The structures and the materials (usually using aluminum for weight reduction with relative strength) are modified for passenger safety. Initially, additional walls were used inside the thin-walled profile, which increased the energy absorption efficiency and the peak value of the crushing force. Another solution to increase energy absorption is to fill the thin-walled profile with a honeycomb structure. However, it increases cost and weight, especially for the honeycomb structures with much thinner walls, and the improvement in energy absorption is limited. Engineers then began to strengthen the structures by filling the interior with lightweight syntactic foams, such as aluminum-silicon carbide syntactic foams [27,28]. The main advantages of such syntactic foams are the ease of filling adaptation to the shape of the thin-walled structures and the possibility of changing the mechanical properties by using balls of different matrix materials of different thicknesses as lightweight porous fillings. In addition, aluminum-ceramic composite foams are widely used as thermal or sound insulation, multifunctional structural materials, in the shipping industry.

However, traditional trial-and-error-based design methods are tedious, ineffective, time-consuming, and even impossible. Biomimicry is an effective and preliminary approach to design porous periodic composites [29,30,31]. In porous material design, biomimetics finds inspiration in natural cellular structures. Over thousands of years of evolution, nature has optimized the microstructural layout of materials. The trick for researchers is to determine how materials can be optimized in specific environments and how to scale up practical engineering applications [32]. Examples of natural periodic cellular materials include nacre, cuttlebone, bamboo, bone, and cork.

Topology optimization (TO) for porous material design is often performed within the finite element analysis (FEA) framework, thereby increasing the level of design capability. TO of isotropic materials is a well-established topic in structural design. Research reports that TO can yield weight savings of up to 30–70% while providing the same or improved functionality. Recently, researchers have started to focus on optimizing the topology of composite materials by tailoring the fiber orientation and constituents, providing innovative tools for robust design [33,34,35,36,37,38,39], which is of high value in the design of advanced material structures.

Characterization of porous or dense composites allows examination of their physical performance. While mechanical tests provide insight into material properties, they are not cost-effective and are time-consuming [40,41,42,43,44]. Alternatively, mathematical models and numerical methods can help characterize such materials [43,45,46,47]. Among them, the homogenization method combined with computer modeling has become popular [21,48,49], where periodic structural representative volume elements (RVEs) are considered and their parameters are regarded as design variables. Thus, relationships between local material parameters (such as density, volume fraction and orientation of reinforcement, and topology of the RVE) and global physical properties (such as Young’s modulus, strength, or conductivity) can be established for structural design and characterization.

In this work, discontinuous carbon fiber-reinforced polymer composites (DiCFRPCs) were used for structural design. Inspired by biomimetics based on the microstructure of cuttlefish bone, in combination with the topology optimization (TO) based on computer modeling was proposed for the development of lightweight multifunctional lattice structural composites. Three-dimensional (3D) periodic lattice blocks were initially designed, inspired by the microstructures of cuttlefish bone. Through TO based on computer modeling, the complex topological structures of the 3D periodic lattice blocks were optimized and realized, and the mechanical properties of the topology-optimized lattice structures were characterized by computer modeling. The results from this work were compared with other similar designs and experiments to validate the effectiveness of the proposed method.

## 2. Topology Optimization of Lattice RVE of DiCFRPCs

TO is a mathematical method that seeks to find the optimal distribution of material within a design domain by minimizing a cost function subject to a set of constraints. It is a powerful tool for engineers and scientists to deliver innovative and high-performance conceptual designs early in the design process, without assuming any prior structural configuration. Therefore, it is suitable for a wide range of applications. Several TO methods exist that use different representations to describe the shapes they refer to. Density-based TO operates on a fixed mesh of finite elements, penalizing the mechanical properties of elements. This penalty uses an interpolation function to find optimal void/solid material distribution that minimizes the objective function [33,50,51,52].

The material structure in its undeformed reference configuration is prescribed by the shape of Ω ⊂ R^d^ (d = 2 or 3 in practice), that is, a bounded, Lipschitz domain whose boundaries decompose into three disjoint parts: ∂Ω = Γ_u_ ∪ Γf ∪ Γ_0_, where Γ_u_ is the part of the specified displacement boundary, Γf is the part of the specified traction force boundary, and Γ_0_ is the part of the traction-free boundary.

For a given design domain (volume) under certain boundary conditions, constraints and loads, the principle of the TO algorithm in a FEA program is to minimize the structural compliance of the domain (maximize the stiffness of the domain) while satisfying the constraints of the structural volume reduction, the spatial material distribution function is used as an optimization parameter [33,53]. The objective function of the TO is to satisfy the structural strain energy under the structural constraints. Reducing the structural strain energy or minimizing the objective function means increasing the stiffness of the structure, and density can be used as a design variable.

For manufacturable results, it is desirable that the solution consist only of solid or hollow elements. To approximate this behavior, intermediate densities are penalized, i.e., intermediate densities are more expensive compared to relative stiffness. This will make the intermediate densities unfavorable. Without this loss, the stiffness-material cost relationship is linear. A popular way to achieve a penalized intermediate density is to express the stiffness of the material as
(1)$$E=\rho^{p}E^{0},\;p>1$$
where **E^0^** is the elasticity tensor of the solid, *ρ* is the density, 0 ≤ *ρ* ≤ 1, and *p* is the penalty index. If the TO procedure deforms the domain in its elastic region, the classical TO problem of minimizing the compliance while constraining the mass using the density method can be expressed as
(2)$$\left\{\begin{array}{c} {min}_{\rho }C\left(\rho \right)=F^{T}u\left(\rho \right)\\ s.t.\left\{\begin{array}{c} \int_{V}^{}\rho dV\leq \alpha V\\ 0\leq \rho_{e}\leq 1,e=1,2,\cdots ,n \end{array}\right. \end{array}\right.$$
where *ρ*_e_ = [*ρ*_1_, *ρ*_2_, …, *ρ*_n_]^T^ is the density vector of the elements, α is the volume reduction ratio, and V is the design volumetric domain before TO operation. The displacements are easily to find using
(3)$$u(\rho )=K^{-1}\left(\rho \right)F$$
where **K** (*ρ*) is the stiffness matrix and ***F*** is the force vector. The density method does not require much extra computing memory, requires only one variable per element, that is the element’s density, and can use any combination of design constraints.

It is assumed that each element containing the matrix and fiber reinforcement can be simplified to an element with a transversely isotropic material in the elemental local coordinate system and the fibers oriented along the local *Z*-axis. The 3D Hooke’s law (stress–strain relationship within the elastic region) for such transversely isotropic material can be written as [54,55,56]
(4)$$\left\{\begin{array}{c} \varepsilon_{x}\\ \varepsilon_{y}\\ \begin{array}{c} \varepsilon_{z}\\ \gamma_{yz}\\ \begin{array}{c} \gamma_{zx}\\ \gamma_{xy} \end{array} \end{array} \end{array}\right\}=\left[\begin{array}{cccccc} \frac{1}{E_{x}} & -\frac{\nu_{xy}}{E_{x}} & -\frac{\nu_{xz}}{E_{x}} & 0 & 0 & 0\\ -\frac{\nu_{yx}}{E_{x}} & \frac{1}{E_{x}} & -\frac{\nu_{yz}}{E_{x}} & 0 & 0 & 0\\ -\frac{\nu_{zx}}{E_{z}} & -\frac{\nu_{zy}}{E_{z}} & \frac{1}{E_{z}} & 0 & 0 & 0\\ 0 & 0 & 0 & \frac{1}{G_{xz}} & 0 & 0\\ 0 & 0 & 0 & 0 & \frac{1}{G_{xz}} & 0\\ 0 & 0 & 0 & 0 & 0 & \frac{1}{G_{xy}} \end{array}\right]\left\{\begin{array}{c} \sigma_{x}\\ \sigma_{y}\\ \begin{array}{c} \sigma_{z}\\ \tau_{yz}\\ \begin{array}{c} \tau_{zx}\\ \tau_{xy} \end{array} \end{array} \end{array}\right\}$$
where the *Z*-axis is the fiber direction, the *x*-*y* plane is the transverse basal plane, and the above compliance matrix is a symmetric matrix; therefore, *E_x_* = *E_y_*, *ν_xy_* = *ν_yx_*, *ν_xz_* = *ν_zx_*, *ν_yz_* = *ν_zy_*, $\frac{\upsilon_{xy}}{E_{x}}=\frac{\upsilon_{yx}}{E_{y}}$, $\frac{\upsilon_{xz}}{E_{x}}=\frac{\upsilon_{zx}}{E_{z}}$, and $G_{xy}=\frac{E_{xy}}{2\left(1+\upsilon_{xy}\right)}$.

A general approach for the biomimetic design and TO of lightweight FRPC lattice structures is shown in the flowchart in Figure 1. First, the FRPC materials should be synthesized and prepared, and the material properties should be characterized. Then, the bio-inspired lattice block as an RVE, with appropriate constraints and boundary conditions, will be topologically optimized until the RVE design satisfies the design criteria. Moreover, the topology-optimized lattice blocks will be prototyped to test or will be numerically tested through computer modeling for its performance and then revised if needed until the design is finalized according to the application requirements. Finally, the finalized design will be manufactured for applications.

As a case study, a DiCFRPC, polyamide 66 with 30 vol.% discontinuous carbon fibers, was chosen for the porous lattice structure design. The cellular structure is anisotropic due to the directional fiber reinforcement and heterogeneous structure in nature. The orthotropic material properties (measured in the elemental local coordinate system) are listed in Table 1 [57].

The commercial FEA software ANSYS was adopted for TO [58]. The 3D 20-node (with higher order shape functions) element type of SOLID186 (current version, corresponding to SOLID95 for the older version of ANSYS) was used for the TO calculations.

Before TO calculation, selecting an initial lattice block and setting initial constraints and boundary conditions according to the biomimetics are critical steps in the design. After carefully observing the transverse cross-sectional profile of the cuttlefish bone and its conditions during the millennium evolution, it was found that for the smaller cuttlefish bone samples with a macroscopic length of about 100 mm, the microstructural lamella spacing is around 100–200 μm in height and pillar spacing is around 80–100 μm in width, which results in lattice blocks with possible aspect ratios between 1 and 2.5. Therefore, an initial 3D periodic block was designed, as shown in Figure 2. In the model, the height is 150 μm, both width and depth are 100 μm, and the ratio of height to width and depth (H:W:D) of 1.5:1:1 was chosen. The top surface is subjected to a pressure of up to 6 MPa, which is associated with the most extreme pressure experienced by the cuttlefish in deep water. The non-topology optimization layer thickness is 8 μm. The sides and corners of the bottom face were constrained, so when the periodic boundary conditions were applied, the periodic block must always uniformly expand laterally during compressive deformation. The elements in the top and bottom layers in grey were not subjected to TO. The initial orientations of the DiCFs in the model were vertically aligned. 

## 3. Results and Discussion

### 3.1. Mesh Convergence Study of the Initial Periodic Lattice Block Model

Before conducting TO calculation, firstly, in order to improve the modeling accuracy, the mesh density or the mesh size of the model needs to be checked and refined. By studying the convergence of the maximum displacement of the model under a pressure on the top surface, as shown in Figure 2, the appropriate mesh size and density were determined to make the modeling results independent of the mesh size or mesh density, as shown in Table 2. It can be seen that mesh convergence is easily achieved with even fewer elements or nodes with larger mesh sizes due to appropriate periodic boundary constraints and symmetric boundary conditions applied, and high-order shape functional elements (3D 20-node solid elements) selected. In order to ensure the accuracy of the TO calculation and accurately represent the optimized domain profile, the actual mesh size used in the TO calculation is 0.03 μm, and the total number of elements and nodes generated for the model is 57,800 and 245,105, respectively. 

### 3.2. Topology Optimization

The ANSYS TO module was used to perform TO of various porosities (material volume reduction) on the periodic lattice blocks with an orthotropic material, such as DiC FRPC. The TO domain can be controlled by customizing regions to be optimized and non-optimized by the predefined bottom layer and the top layer considered as non-optimized regions. During the OT process, the objective function (structural compliance) decreases as the number of iterations increases, and after a certain number of iterations, the iterative results finally converge to a stable minimum value, indicating that the structural rigidity reaches its maximum value, and the complex optimized topology is realized. Figure 3 shows that the material on the front of the lattice block represents the progressive iterative modeling process from the beginning to the end of the TO process, where the red color (at the right end of the color bar) represents 100 vol.% material/solid, while the dark blue color (at the left end of the color bar) represents 0 vol.% material/void. It can be seen that the TO modeling iterative process is rapidly approaching the convergence state, and after the entire TO process reaches 40%, the modeling results are very similar to those at the end of the TO process. Figure 4 shows a complex topology-optimized lattice block with 90% porosity (90% material volume reduction) of the initial 3D periodic block. The surface profile shown in the figure was smoothed based on the surface profile represented by the elements using SolidWorks, a 3D solid modeling computer-aided design and computer-aided engineering software. Comparing the optimized block with the smaller cuttlefish bone unit-cell structures having similar porosity [21], the two structural topologies look similar.

### 3.3. Mechanical Property Characterization of the Topology-Optimized Lattice Block

After achieving the topology-optimized lattice block structures for various porosities, the mechanical properties of the topology-optimized periodic lattice blocks should be characterized. As a case study, compression tests were carried out through FEA modeling using ANSYS software. The applied loads, the boundary conditions, and the constraints are similar to the mesh convergence tests, referring to Figure 2. A pressure was applied on the top surface (Z = height) of the optimized lattice block. The bottom surface (Z = 0 and U_Z_ = 0), the back face (X = 0 and U_X_ = 0), and the left face (Y = 0 and U_Y_ = 0) are set as symmetric faces. The front face (X = depth and the displacements in the X-direction are the same for the entire face) and the right face (Y = width and the displacements in the Y-direction are the same for the entire face) are set as periodic faces. Before material property characterization by FEA modeling, for each lattice block model with a specific porosity, a mesh convergence study was first carried out to determine the appropriate mesh size. Table 3 lists the mesh convergence study data for the 90% porosity topology-optimized lattice block model, in which the maximum displacement and Young’s modulus were recorded and extracted for mesh determination criteria. Since the models are topologically irregular, meshing the model with hexahedral elements would cause algorithmic difficulties, therefore, the tetrahedral elements were used. After mesh convergence study, the 90% porosity lattice block model and other porosity lattice block models all adopted a mesh size of 3 μm, which is consistent with the mesh size used in the initial lattice block model. Since the maximum stress is usually induced at the edges or corners or notches of the model and is very sensitive to the mesh size, it is generally not recommended as a criterion for mesh convergence studies. The different specific Young’s moduli in the Z-direction (fiber-direction) versus the porosity of the topology-optimized lattice blocks are shown in Figure 5. It can be seen that these relationships are generally not a monotonically changing pattern (monotonically increasing or monotonically decreasing). There may be an inflection point where the material performance (properties) is at a maximum or minimum point, which can be adequately accounted for when effectively designing novel material lattice blocks.

The partially assembled porous structure formed by the topology-optimized periodic lattice blocks is shown in Figure 6a, where the length is four cells, the depth is four cells, and the height is three cells. Comparing the cross-sectional structural profile of the cuttlefish bone [21], it is easy to find the structural similarity between the actual model and the topology-optimized model. The color plot on the partially assembled structure shows the vertical compressive stress distribution, with higher stress at the bottom of each pillar and lower stress at the upper structure of each cell. Figure 6b shows the physically assembled structure by 3D printing using plastic (ABS), with a length of two cells, a depth of two cells, and a height of two cells. The topology-optimized model was converted into a 3D digital model using SolidWorks and then input into the 3D printer. The 3D printing process of such complex geometric structures is time-consuming.

### 3.4. Validations

To better evaluate the outcomes from this work and to validate the effectiveness of the current work, the current work is compared with the work from ref. [59] and the results are listed in Table 4 and plotted in the format of Ashby chart in Figure 7 [60]. It provides a normalized platform for validating the current work by comparing some of the material properties related to density, porosity, matrix and reinforcement, Young’s modulus, Poisson’s ratio, and their specific properties relative to their density. The materials include the biomimetic designed and topology-optimized 3D lattice structural material (this work), many common bulk materials, and porous FRPCs. According to the research in ref. [59], the materials used to fabricate the samples are PLA and DiCFR PLA. The RVE models designed from the reference were based on the biomimetic inspiration without TO simulation, such as 2D hexagon model, 2D cuttlefish bone model, and 3D octahedron model. According to the relevant standard test methods for the tensile properties of polymer-based composites provided by the American Society for Testing and Materials (ASTM), the testing compressive loads were applied in the X, Y, or Z directions, respectively. The samples from the reference were prepared by 3D printing process, and as the DiCFRPLA composite filament used in 3D printing, the discontinuous fibers were mainly aligned along the printing path. Therefore, the printing direction is basically the fiber direction for the samples.

From the data in the chart (Figure 7) and Table 4, and the research work in the reference, it was found that the printing directions (also referring to the fiber directions) for DiCFRC, load types (referring to tension or compression), and porosity (volumetric reduction) have great effects on the samples’ stiffnesses (Young’s modulus) and specific Young’s modulus. It is also clear that the specific Young’s moduli (E/ρ) of the 2D and 3D cuttlefish bone models are higher than or equivalent to that of the CFRPLA bulk composites. The 3D octahedron models may be optimized for better performance. However, this research shows that the specific Young’s modulus of the biomimetic-inspired and TO lattice structures have much higher values, which validated the effectiveness of the current design approach for the lattice structural design and also explains why the cuttlefishes can withstand high pressure in the deep sea.

## 4. Conclusions

In this work, a biomimetic design inspired by the microstructure of cuttlefish bone and TO of lightweight multifunctional periodic lattice blocks were performed. The material used for the structural design is discontinuous carbon fiber-reinforced nylon material composites. The structural compliance was minimized and the structural stiffness was maximized through TO of the 3D periodic lattice blocks based on FEA modeling. Through 3D TO, the complex topologies of the 3D periodic lattice blocks were realized, and the mechanical properties of the topology-optimized lattice structures were characterized by computer modeling. The results of this work were compared with other similar designs and experiments to validate the effectiveness of the proposed approach. The proposed approach provides a design tool for developing more affordable and higher-performance structural materials.

Further studies can be performed based on various types of biomaterial structures (such as various fibers or various polymeric matrices) with various dimensions of periodic lattice blocks under various constraints or boundary conditions for desired applications.

## Figures and Tables

**Figure 1 biomimetics-08-00148-f001:**
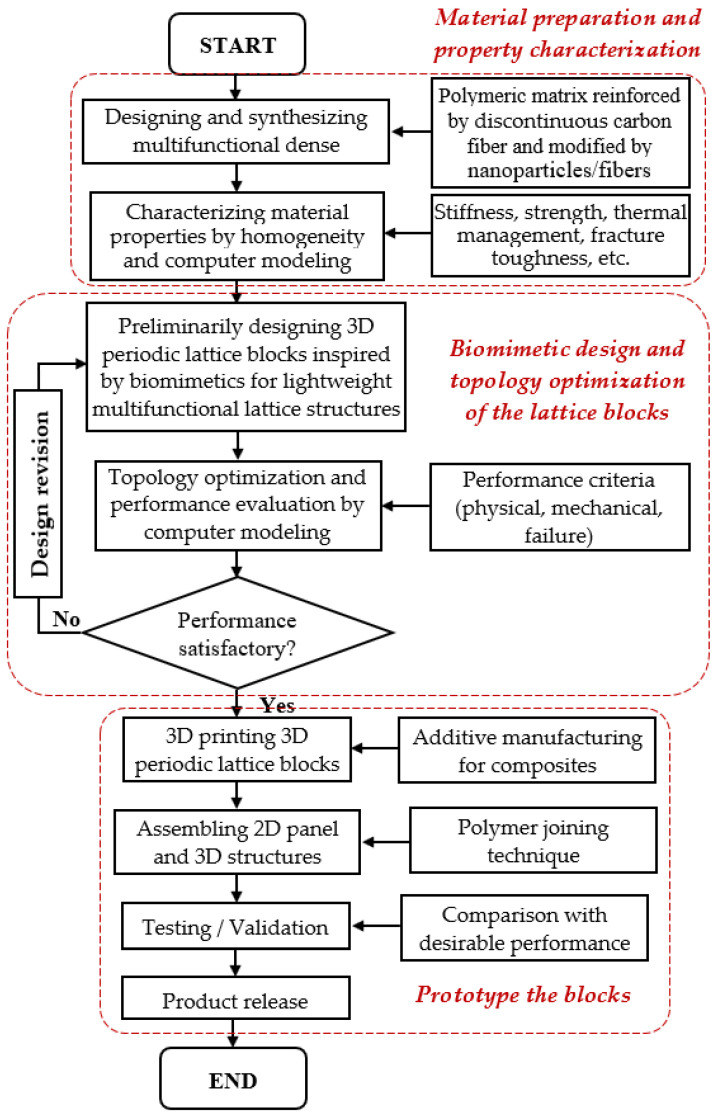
Flowchart of the biomimetic design, topology optimization, and fabrication of lightweight fiber-reinforced polymer composite lattice structures.

**Figure 2 biomimetics-08-00148-f002:**
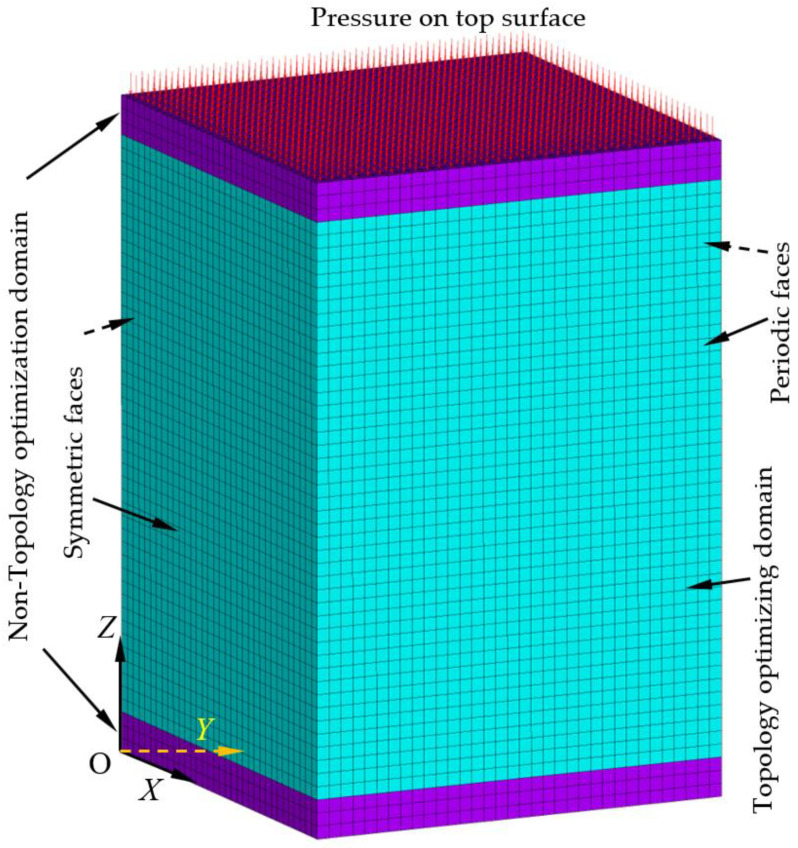
Initial 3D solid periodic lattice block (topological design domain) with pressure on the top and periodic boundary constraints and surrounding symmetric boundary conditions.

**Figure 3 biomimetics-08-00148-f003:**
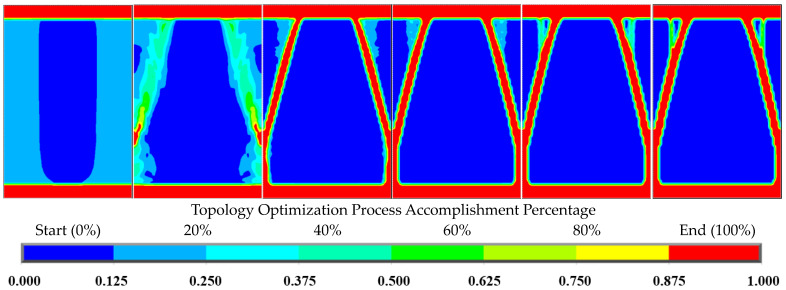
The progressive material representation of the lattice block front face through the TO process.

**Figure 4 biomimetics-08-00148-f004:**
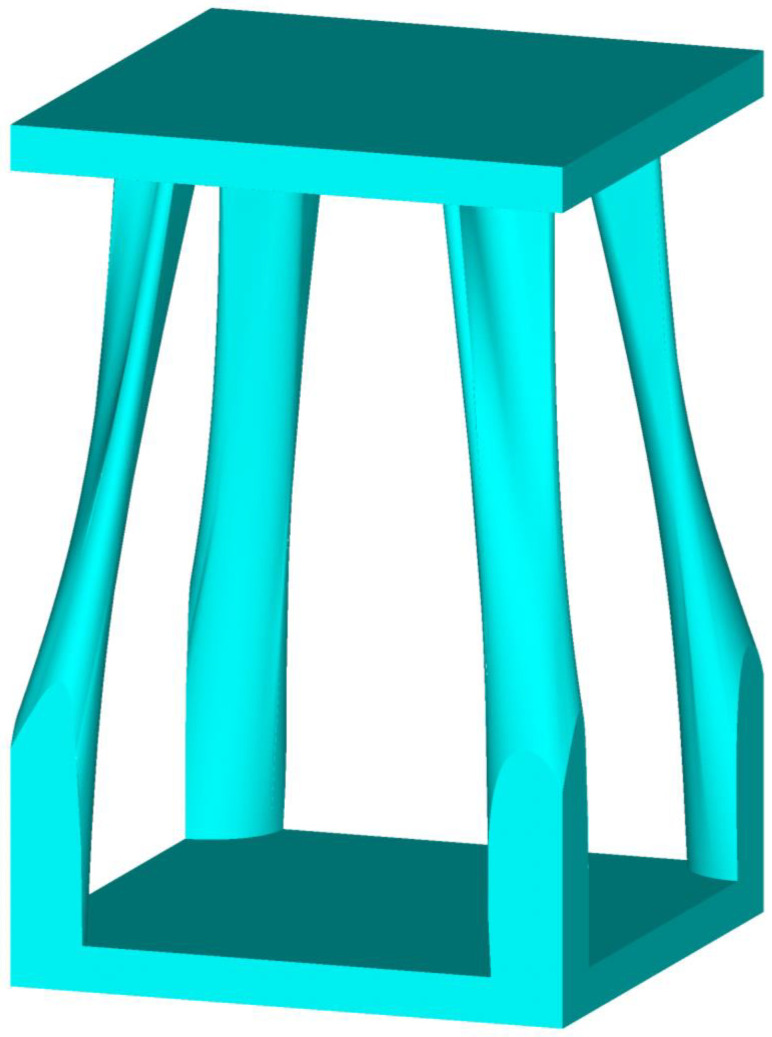
Optimized 3D periodic lattice block with a porosity of 90% of the TO domain.

**Figure 5 biomimetics-08-00148-f005:**
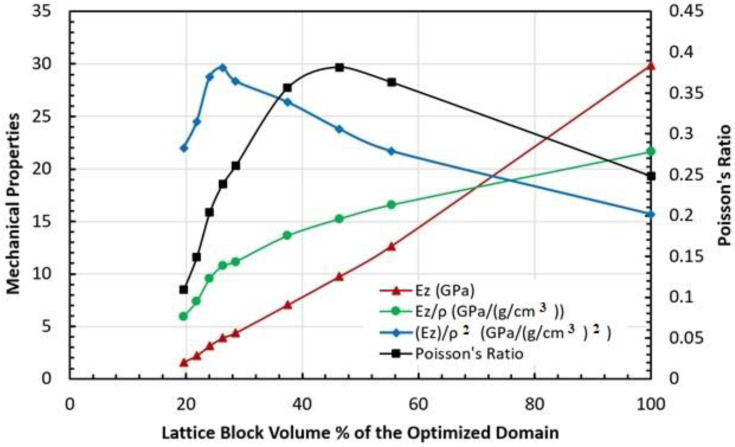
Relationships of some mechanical properties in Z-direction (fiber-direction) vs. volume percentage of the topology-optimized lattice blocks.

**Figure 6 biomimetics-08-00148-f006:**
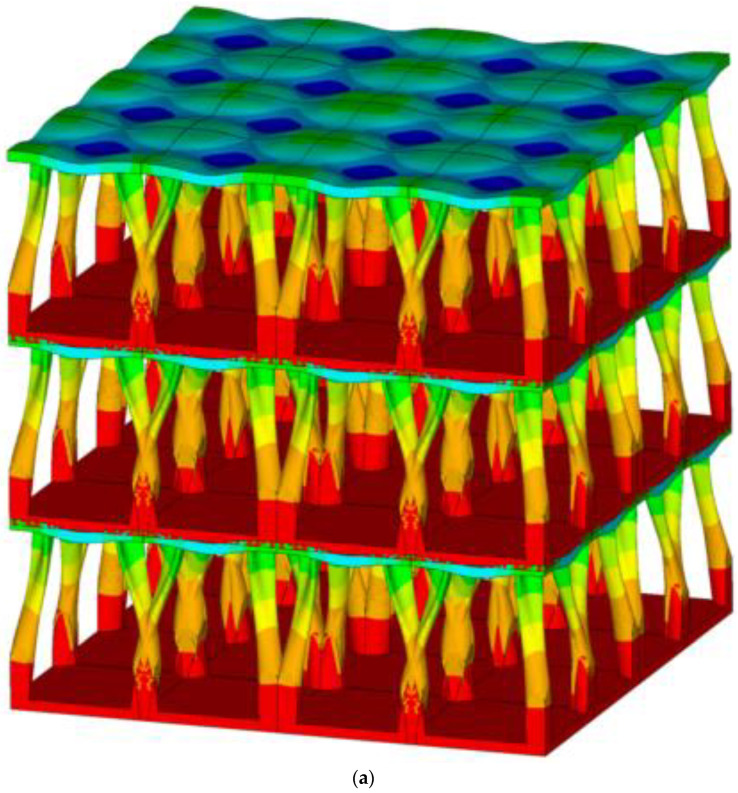
A partially assembled porous structure (**a**) formed by the topology-optimized periodic lattice blocks (4 cells in length, 4 cells in depth, and 3 cells in height) and (**b**) fabricated by plastic (ABS) 3D printing (2 cells in length, 2 cells in depth, and 2 cells in height).

**Figure 7 biomimetics-08-00148-f007:**
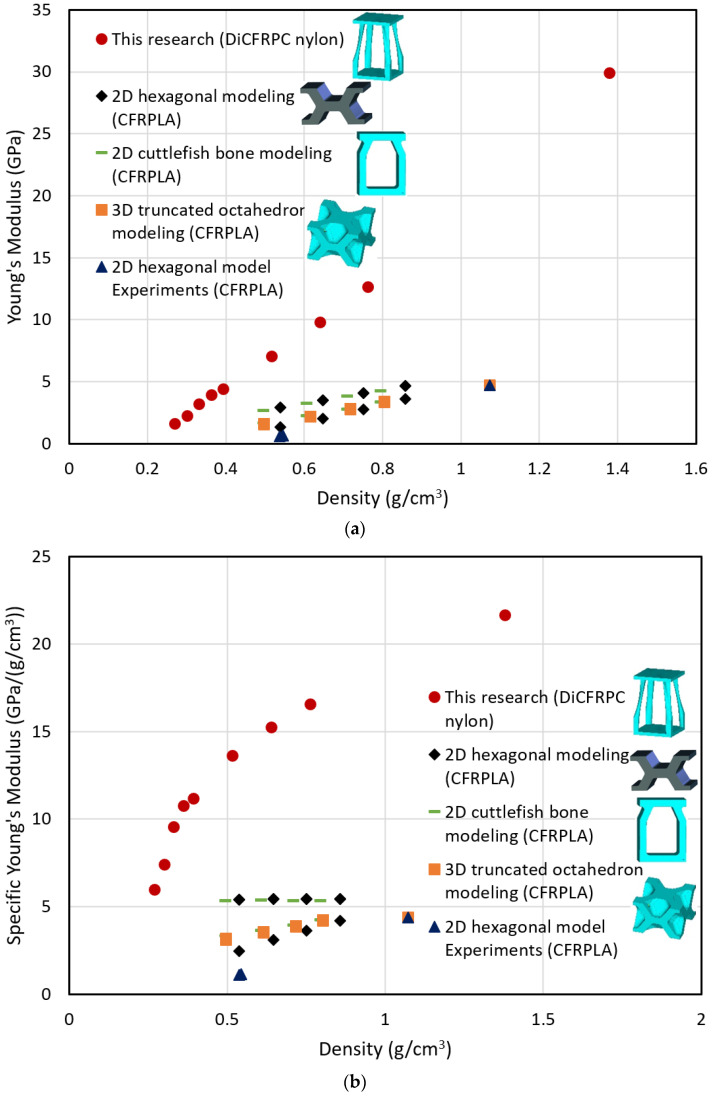
Comparison of (**a**) Young’s modulus vs. density and (**b**) specific Young’s modulus vs. density from the current research and the research results from ref. [59].

**Table 1 biomimetics-08-00148-t001:** Material orthotropic elastic properties of the DiCFRPC lattice structural blocks (*Z*-axis is the fiber orientation).

Young’s Modulus (GPa)		Shear Modulus (GPa)		Poisson’s Ratio		Density *ρ* (kg/m^3^)
*E_x_*	5.57	*G_xy_*	2.16	*ν_xy_*	0.29	1380
*E_y_*	5.57	*G_yz_*	7.41	*ν_yz_*	0.047	
*E_z_*	29.9	*G_zx_*	7.41	*ν_zx_*	0.25	

**Table 2 biomimetics-08-00148-t002:** Mesh convergence study of the initially designed lattice block under a pressure on the top surface with periodic boundary constraints and symmetric boundary conditions.

Mesh Size (μm)	Total Elements	Total Nodes	Maximum Displacement (μm)
50	12	111	0.0309
40	36	264	0.0309
35	45	320	0.0309
30	80	515	0.0309
25	96	605	0.0309
20	200	1152	0.0309
17.5	324	1771	0.0309
15	490	2576	0.0309
12.5	768	3897	0.0309
10	1500	7271	0.0309
8	3211	14,924	0.0309
6	7225	32,436	0.0309
5	12,000	52,940	0.0309
4	23,750	102,752	0.0309
3	57,800	245,105	0.0309

**Table 3 biomimetics-08-00148-t003:** Mesh convergence study of the 90% porosity topology-optimized lattice block under a pressure on the top surface with periodic boundary constraints and symmetric boundary conditions.

Mesh Size (μm)	Total Elements	Total Nodes	Maximum Displacement (μm)	Young’s Modulus (GPa)
10	85,097	129,527	9.1033	1.0471
5	124,364	185,968	9.1781	1.0716
4	137,916	206,287	9.1815	1.0732
3	183,201	273,919	9.1904	1.0732

**Table 4 biomimetics-08-00148-t004:** Comparison summary of the properties for some biomimetic designed structures (*Z*-axis is the fiber direction for this work, but the printing paths were not always along the *Z*-axis in ref. [59]).

Material	Density ρ (g/cm^3^)	Volume %	E (GPa)	E/ρ (10^6^ m^2^ S^−2^)	νE/ρ (10^6^ m^2^ S^−2^)
PLA	1.19	100	2.865	2.408	1.422
CFRPLA *	1.073	100	4.711	4.390	2.023
2D Hexagon PLA in X-lateral **	0.609	51.2	0.505	0.829	1.167
2D Hexagon PLA in Y-lateral **	0.626	52.6	0.455	0.727	1.078
2D Hexagon ** CFRPLA in X-lateral	0.545	50.8	0.622	1.141	1.447
2D Hexagon ** CFRPLA in lateral	0.537	50.0	0.588	1.096	1.428
2D Hexagon ** CFRPLA in Y-lateral	0.545	50.8	0.622	1.141	1.447
2D Cuttlebone PLA ** in X-lateral	0.610	51.3	0.829	1.358	1.493
2D Cuttlebone PLA ** in Y-lateral	0.610	51.3	1.470	2.408	1.988
2D Cuttlebone ** CFRPLA in X-lateral	0.610	51.3	1.926	3.155	2.275
2D Cuttlebone ** CFRPLA in Y-lateral	0.610	51.3	2.939	4.814	2.810
3D Octahedron ** PLA in *X*-axis	0.617	51.9	0.923	1.496	1.557
3D Octahedron ** PLA in *Y*-axis	0.617	51.9	0.853	1.383	1.497
3D Octahedron ** PLA in *Z*-axis	0.617	51.9	0.943	1.529	1.574
3D Octahedron ** CFRPLA in *X*-axis	0.556	51.9	1.853	3.331	2.448
3D Octahedron ** CFRPLA in *Y*-axis	0.556	51.9	1.682	3.023	2.333
3D Octahedron ** CFRPLA in *Z*-axis	0.556	51.9	1.880	3.379	2.466
Aluminum	~2.710	100	~69.0	~26.0	3.077
Steel	~7.85	100	~200	~25.0	1.790
Titanium alloys	~4.50	100	~112.5	~25.0	25.00
Diamond (C)	~3.53	100	~1220	~346	9.895
This research (Nylon 66 with 30 vol.% CFs) in *Z*-axis	1.380	100	29.87	21.64	3.960
This research (Nylon 66 with 30 vol.% CFs) in *Z*-axis	0.7636	55.3	12.65	16.57	4.658

Note: * Carbon fiber average diameter of 7 μm, average length of 150 μm, and aspect ratio of 21.4; 15 vol.% CFs for CFRPLA. ** Experimental results.

## Data Availability

Not applicable.

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
