# Peer review of "Biomimetic Design and Topology Optimization of Discontinuous Carbon Fiber-Reinforced Composite Lattice Structures"

_biomimetics, 2023, doi:10.3390/biomimetics8020148_

Round 1

Reviewer 1 Report

The subject of the study is numerical analysis of bio-inspired porous structure. The conditions of the analysis carried out consisted of surface pressure loading of the structure and observation of its behavior. The study of the porous composite structure was inspired by the cuttlefish tail, while the main goal was to optimize its mechanical properties. The analysis was based on topology optimization, the schematic diagram of which is shown in Figure 1.

 The subject of the research is described quite extensively, moreover, supported by relevant graphics, it forms a whole, thoroughly explaining the investigated models. The obtained conclusions are presented in a logical and coherent manner for the reader. The methodology of the research conducted, as well as the results and conclusions, are correctly stated, and the work itself is of high scientific value, however some parts of the proposed manuscript need improvement:

-         Figure 6 lacks a legend to correctly analyze the graphic

-    Too short text referring to table 3. The table presented contains a large amount of data, no detailed reference of the studied structure to other materials. The current form looks as if table 3 is a filler of the article, this should be corrected

-    The analysis lacks several relevant literature items, making the literature review and conclusions more relevant e.g., the paragraph on porous structures (lines 72-77) reinforced with composite particles lacks a description of porous structures reinforced with silicon carbide particles (sample publications below)

DOI: 10.1016/j.compstruct.2022.116102

           DOI: 10.12913/22998624/153028

Author Response

Dear Reviewers,

Many thanks for your taking time to review my manuscript and provide me with your valuable comments. Attached please find my responses. Please feel free to contact me if you have questions or need further information from me.

Regards,

Reviewer 2 Report

This manuscript presents a biomimetic lattice design inspired by cuttlefish bone. A topological optimization using computer modeling was performed to explore the best solution with minimized structural compliance and maximized structural stiffness for 3D periodic discontinuous carbon fiber-reinforced nylon lattice blocks. The mechanical properties of the topologically optimized lattice structures were characterized by computer modeling and were compared with previously reported similar designs, demonstrating the effectiveness of the proposed approach.

This manuscript shows good organization but falls short of clarity and rigor. I do not recommend this manuscript for publication before the author could consider the following questions and revise the manuscript. Detailed comments are listed below.

1.       In general, I have no doubt about the section arrangement. However, there is an unacceptable skip between the display of biological cuttlefish bone (Fig 2) and the topologically optimized design (Fig 4). It is unclear how the author reaches the design in Fig 4 from the initial block structure. Since there is a number of iterations, it is better to demonstrate how these iterations provide us with gradually improved structures using an extra figure.

2.       Mesh convergence study was conducted to decide on a mesh size of 0.03 um in the actual study. It should be double-checked whether this number is correct, given the minimum displayed mesh size as 4 um in Table 2. Moreover, in addition to evaluating the influence of mesh size on maximum displacement, I would like to see how it affects the mechanical properties the author currently presents in Fig 5. It is more convincing if the mechanical properties converge as the mesh size gets smaller.

3.       Fig 6: what is its dimensional/mechanical/structural information?

4.       The visualization of table 3 is not satisfactory. In addition to the ambiguous difference between the last two rows, it is better to use an Ashby chart plot to replace table 3, like Fig 4 in the following paper: Zheng, Xiaoyu, et al. "Ultralight, ultrastiff mechanical metamaterials." Science 344.6190 (2014): 1373-1377.

5.       Mechanical property characterizations were only performed using FEA, which is not following the flowchart in Fig 1. Experiments are essential to examining the FEA results. Why did the author not conduct 3D printing and experiments?

Author Response

(The authors gave the same response as above.)

Round 2

Reviewer 2 Report

The author addresses my questions, and I don't have further concerns. The manuscript could be accepted in its current form.